# Structural Analysis of the *A* Mating Type Locus and Development of the Mating Type Marker of *Agaricus bisporus* var. *bisporus*

**DOI:** 10.3390/jof9030284

**Published:** 2023-02-21

**Authors:** Yeon-Jae Choi, Sujin Jung, Hyerang Eom, Thimen Hoang, Hui-Gang Han, Sinil Kim, Hyeon-Su Ro

**Affiliations:** Department of BioMedical Bigdata (BK21), Research Institute of Life Sciences, Gyeongsang National University, Jinju 52828, Republic of Korea

**Keywords:** *Agaricus bisporus*, *A* mating type, transposon, nuclear marker

## Abstract

Karyotyping in *Agaricus bisporus* is crucial for both the isolation of homokaryotic strains and the confirmation of dikaryon establishment. For the verification of the karyotype, the *A* mating type loci of two homokaryotic strains, H39 and H97, were analyzed through comparative sequence analysis. The two loci showed major differences in two sequence regions designated as Region 1 and Region 2. H97 had a putative DNA transposon in Region 1 that had target site duplications (TSDs), terminal inverted repeats (TIRs), and a loop sequence, in contrast to H39, which only had the insertional target sequence. Homologous sequences of the transposon were discovered in the two different chromosomes of H97 and in one of H39, all of which have different TSDs but share high sequence homology in TIR. Region 2 shared three consensus sequences between H97 and H39. However, it was only from H97 that a large insertional sequence of unknown origin was discovered between the first and second consensus sequences. The difference in length in Region 1, employed for the verification of the *A* mating type, resulted in the successful verification of mating types in the heterokaryotic and homokaryotic strains. This length difference enables the discrimination between homo- and heterokaryotic spores by PCR. The present study suggests that the *A* mating type locus in *A. bisporus* H97 has evolved through transposon insertion, allowing the discrimination of the mating type, and thus the nuclear type, between *A. bisporus* H97 and H39.

## 1. Introduction

*Agaricus bisporus* var. *bisporus* (abbreviated as *A. bisporus*) is the most commonly cultivated edible mushroom in the world. Because of its economic importance, many cultivars are being developed, and in recent years, there has been increasing demand for the development of new varieties in response to environmental changes, such as global warming, to ensure the sustainability of the mushroom industry. In the generation of new strains, the traditional cross-mating between two different monokaryons has been the major technique employed in mushroom breeding. Molecular breeding by modifying genomic sequences to alter the function and expression of genes of interest [1] or by introducing foreign DNA to the host chromosomal sequence [2,3,4] has been attempted with some mushrooms; however, this approach is not applicable to edible mushrooms because the mushrooms generated in this way are essentially genetically modified organisms (GMOs). Nonetheless, a new technology in molecular breeding has emerged recently, namely CRISPR-Cas9 gene-editing technology, which enables targeted modification of the genes or DNA sequences of certain functions [5,6,7,8]. It has been applied to generate a new strain of *A. bisporus* that is resistant to browning via gene-editing of polyphenol oxidase (PPO) [9]. PPO-modified *A. bisporus* was the first gene-edited organism to circumvent the US FDA’s GMO regulation, opening a new avenue for the breeding of mushrooms.

*A. bisporus* is a pseudo-homothallic species of *Agaricus* that primarily produces bi-nucleated spores while also producing a minor portion of mononuclear spores. The bi-nucleated spores develop into self-fertilizing heterokaryotic mycelia (n + n), which are capable of producing fruiting bodies without the need for mating [10]. However, homokaryotic mycelia (n), which are generated when mononuclear spores germinate, must be mated with a suitable mating partner of a different mating type for the completion of sexual reproduction. Because of its unique life cycle, breeding *A. bisporus* has a number of issues regardless of whether it involves conventional cross-mating or molecular breeding. Firstly, it is very hard to isolate homokaryotic (n) strains with a desirable genetic trait from the basidiospores because the spore-derived mycelia are essentially heterokaryotic (n + n), even without mating [10]. Studies on basidiospores have shown that the rate of homokaryons versus heterokaryons is less than 3% [10,11,12]. Secondly, it is impossible to discriminate the mated heterokaryotic mycelia from the homokaryotic mycelia using microscopic observation, which has been a common practice in mushroom breeding, because *A. bisporus* lack clamp connections on the heterokaryotic mycelia. Moreover, *A. bisporus* can have up to 25 nuclei in a mycelial cell [13], presenting challenges particularly in molecular breeding where a modified nucleus has to be selected from the multiple unmodified nuclei within the targeted mycelial cell. Dedikaryotization provides a useful solution to this issue, allowing for the generation of homokaryons of *A. bisporus* through a simple process that involves plating protoplasts derived from heterokaryotic mycelium and selecting for homokaryotic regenerants [14]. Nevertheless, the generation and verification of homokaryons are still some of the main hurdles in the breeding of *A. bisporus*.

Verification of the nucleus in *A. bisporus* has been approached by various fingerprinting methods, including restriction fragment length polymorphisms (RFLPs) [12], inter-simple sequence repeats (ISSRs) [11,15], and short sequence repeats (SSRs) [16]. However, all of these have some degree of weakness in reproducibility and in practical application. Alternatively, sequence variations in the allelic gene can be markers for each nucleus in the dikaryotic cell [17,18,19]. In particular, sequence regions in the *A* mating type locus have become successful nuclear markers for *Lentinula edodes* [20] and *Pleurotus eryngii* [21]. The *A* mating type locus plays a critical role in the recognition of compatible nuclei for mating partners [22].

In the present study, we analyzed the *A* mating type loci of two homokaryotic strains (H97 and H39) of *A. bisporus,* which were the parental strains of the heterokaryotic Horst U1 cultivar [23,24], to further understand structural differences and to develop nuclear markers. Our investigation revealed that insertion of two DNA fragments differentiated the mating type locus of H39 and H97 and thereby enabled PCR-based verification of the *A* mating type by targeting the insertional sequences.

## 2. Materials and Methods

### 2.1. Strains and Culture Conditions

The strains of *A. bisporus,* including the heterokaryotic strains (1246 and 1225) and the homokaryotic strains (1062, 1246-91, and 1225-78), were donated by the mushroom stock center of the National Institute of Horticultural and Herbal Science, RDA, Korea. The heterokaryotic strains, including NH1, Top, and EM, were isolated directly from the fruiting bodies obtained from commercial markets. The mycelial culture was conducted in a compost–dextrose–peptone medium (CDPM) containing 1% glucose, 0.7% malt extract, and 0.5% peptone in compost extract, as previously reported [4]. The mycelia were cultivated in CDPM at 25 °C. 

### 2.2. Sequence Analysis of the A Mating Type Locus

Sequence information for the *A* mating type locus located at chromosome 1 (GenBank Acc. Nos. CP015470.1 for H39 and CP015457.1 for H97) of *A. bisporus* was retrieved from GenBank. Gene arrangement in the *A* mating type locus of H39 was assigned manually based on that of H97. Inverted repeats were predicted by the einverted program provided by EMBOSS explorer (http://emboss.toulouse.inra.fr/cgi-bin/emboss/einverted), accessed on 23 November 2022. Secondary structure prediction of the inserted sequence found in the *A* mating type locus of H97 was performed using the RNAfold web server (http://rna.tbi.univie.ac.at/cgi-bin/RNAWebSuite/RNAfold.cgi), accessed on 12 January 2023.

### 2.3. Genomic DNA Extraction

Genomic DNA was extracted from the mycelia of *A. bisporus* cultured in the CPDM medium for two weeks at 25 °C. In brief, the mycelia were collected by centrifugation at 3000× *g* for 10 min. The harvested mycelia were frozen in liquid nitrogen and then ground using a mortar and pestle. Total genomic DNA was isolated from the ground mycelial powder using a genomic DNA prep kit (HiGene™; BIOFACT, Daejeon, Republic of Korea). 

### 2.4. Identification of Mating Types

Identification of the mating type was achieved using a targeted PCR with a primer set (5′-ATCAGTACTTTGAAGCGTGGTAG-3′ and 5′-GTGCTCATACCTCACAATCACTG-3′) specific to the variable sequence in Region 1 of the *A* mating type locus. The PCR was conducted in a reaction mixture containing a template DNA (200 ng), 2X dye mix (Enzynomix, Deajeon, Republic of Korea), and 1 μM of each primer under the following conditions: initial denaturation at 95 °C for 5 min; 28 cycles of denaturation at 95 °C for 30 s, annealing at 57 °C for 30 s, and elongation at 72 °C for 2 min; and final elongation at 72 °C for 5 min. 

### 2.5. Isolation of Homokaryotic Strains from Basidiospores 

In order to generate spore-borne mycelia, basidiospores collected from the fruiting bodies of *A. bisporus* NH1 were spread on a CDPM agar plate and then grown at 25 °C. Fast-growing mycelia were removed from the plate as often as possible during the incubation to exclude heterokaryotic single spore cultures, since heterokaryons grow generally faster than homokaryons. Slow-growing colonies were taken from the plate and grown on CDPM agar for further analysis.

## 3. Results

### 3.1. The A Mating Type Locus Structure of A. bisporus H39

The detailed structure of the *A* mating type locus of homokaryotic strain H39, a parental strain to generate the heterokaryotic Horst U1 cultivar [23,24], was investigated through sequence comparison with the *A* mating type locus of another parental homokaryon, H97. The *A* mating type locus of H97 has been reported to consist of two β*-fg* genes, three homeodomain protein (HD)-coding genes (*a1-1*, *a2-1*, and *b2-2*), and *mip1* [19]. The *A* mating type locus of H39 is located at chromosome 1 (spanning 947,581–960,644), the same as that of H97. Both of the *A* mating type loci encode *a1-1* and *a2-1*, which form a heterodimeric transcription factor for mating gene regulation [22], located between β*-fg* and *mip1* (Figure 1). However, it was discovered that the extra copy of *HD* (*b2-2*) in H97 vanished, along with a 3 kb non-coding sequence in H39 (Figure 1A, Region 2). H39 also lacked a sequence region of 1441 bp between β*-fg1* and β*-fg2* (Figure 1A, Region 1). These two losses shortened the *A* mating type locus of H39 by 4214 bp. 

Region 1 in H39 was made up of a serial connection of a sequence unit CTAAAGTCC, a consensus sequence 1 (92 bp), an insertion (177 bp), and a consensus sequence 2 (256 bp). However, in H97, it was composed of two CTAAAGTCC units with a large insertion (1441 bp), a consensus sequence 1 (92 bp), and a consensus sequence 2 (256 bp) (Figure 1B), indicating that Region 1 of H97 was expanded by the 1441 bp insertion situated between the target site duplication (TSD) of CTAAAGTCC, while losing the 177 bp insertion. Region 2 in H39, located between β*-fg*2 and *a2-1*, was a 791 bp sequence with no coding gene inside, whereas the same region in H97 comprised a much longer sequence (3752 bp) with an extra copy of the HD gene (*b2-2*). Sequence comparison of Region 2 revealed that there were three consensus sequences between H39 and H97, of which the first and third occurred immediately after β*-fg*2 and just before *a2-1*, respectively (Figure 1C). The HD gene *b2-2* was incorporated between the first and second consensus sequences in H97, whereas the second consensus sequence was located at a distance of 72 bp from the first consensus sequence in H39 (Figure 1C).

### 3.2. Analysis of Region 1 in the A Mating Type Locus

Further analysis of Region 1 revealed that it contained three inverted repeats (IRs), of which IR1 was located immediately after the 5′-TSD and just before the 3′-TSD, while IR2 occurred at 6 bp after the 5′-side of IR1, and IR3 was connected to the 3′-side of IR1. IR2 and IR3 contained Loop 1 (223 bp) and Loop 2 (972 bp), respectively (Figure 1B and Figure 2A). The sequence arrangement shown in Region 1 of H97, TSD–IR–insertion–IR–TSD, is a typical structure of a DNA transposon, although neither of the loops carry intact transposase genes. A secondary structure prediction by RNAfold resulted in the structure of a transposon with the terminal inverted repeat (TIR, 93 bp), consisting of 5′-TIR by 5′-IR1 and 5′-IR3 and 3′–TIR by 3-’IR3 and 3′-IR1, as shown in Figure 2A. The transposition appears to have occurred at the target site in the ancient chromosome 1, since the same site in H39 did not have this insertion (Figure 1B and Figure 2C). A more intact form of this structure, which lacks Loop 1, was found at chromosome 7 (GenBank Acc. No. CP015463.1: 1242159–1243137) of H97 but with a different TSD (GTCTGGGTG) (Figure 2B). The TIR here was a 96 bp-long sequence highly homologous to the TIR region of the one found at chromosome 1 of H97. H39 did not have the transposon insertion at the target site in chromosome 7, similar to that of chromosome 1 (Figure 2D). The same structures with different TSDs and sequence variations in the TIR were found at chromosome 4 of H39 and H97 (Figure 2E). Other structures with a homologous but shortened TIR (21 bp) were found at chromosome 3 of H39 and H97 and at chromosome 8 of H39 (Figure 2F). The Loop 2 homologues are provided in Appendix A. The Loop 1 incorporated inside the 5′-TIR of Region 1 is presumably another type of transposon that forms a palindromic structure by IR2. Multiple homologous sequences were found in almost every chromosomal DNA of H97 and H39, except for chromosome 11 of H97 and chromosome 4 of H39 (Appendix A). They could be part of the retrotransposon.

### 3.3. Simple Verification of the A Mating Type through PCR

Verification of the mating type is the first step in mushroom breeding. This can be achieved either by cross-mating analysis among the monokaryotic mycelia generated from the basidiospores or by direct sequence analysis of the mating type locus. Different from the tetrapolar mating behavior in most mushrooms, *A. bisporus* mates in a bipolar manner, governed only by the *A* mating type locus [19,25]. Therefore, any sequence variation in the *A* mating type loci of compatible mating pairs can be a marker for the mating type of the monokaryotic strains and a marker for a nucleus in the dikaryotic cytoplasm as well. Accordingly, we assessed the *A* mating type of *A. bisporus* using the length difference in Region 1 (Figure 1), by designating Region 1 in the homokaryotic H39 strain (427 bp) as the *A1* mating type and that in the H97 strain (1699 bp) as the *A2* mating type. The PCR analysis targeting Region 1 resulted in the successful recognition of the two *A* mating types in the heterokaryotic strains as an indication of the presence of two nuclei (Figure 3A). It also showed that the three homokaryotic strains stocked in RDA are the *A1* mating type. The verification method was employed to distinguish the mating type of the basidiospores collected from the fruiting body of *A. bisporus* NH1. The majority of the spore-borne mycelia (82/102) were bi-nucleated and showed the presence of both the *A1* and *A2* mating types, while 20 were homokaryons (Figure 3B). Among the 20 homokaryons, eight were *A1* and 12 were *A2*. The rate of homokaryons in this experiment was 19.6%, which was extraordinarily higher than the previously reported value of less than 3% [11,12]. This potentially came from the biased selection process of spore-borne mycelia that targeted the isolation of slow-growing mycelia. It is known that homokaryons grow slower than heterokaryons [10,26].

### 3.4. Validation of the A Mating Type Marker and Comparison with the 39Tr 2/5-2/4 Marker

Validation of our *A* mating type marker was performed by direct sequencing of the variable sequence region located between the HD domain regions of *a1-1* and *a2-1* in the *A* mating type loci, as previously described [20]. Our sequence analysis on some of the homokaryons in Figure 3B revealed that the *A1* homokaryons shared identical sequences, while the A2 homokaryons shared the same sequence polymorphism, as shown in Figure 4A. Validity of the *A* mating type marker was further demonstrated by the successful mating of Nos. 23 and 24 with No. 13 and failure in mating Nos. 13 and 19, as well as Nos. 22 and 23 (Figure 4B).

The karyotypes of the spore-borne mycelia shown in Figure 3B were compared to those determined using the 39Tr 2/5-2/4 marker, an approach developed by Gao et al. [18]. The 39Tr 2/5-2/4 marker is located on chromosome 7 of H39 (CP015476.1:748009-748677, 668 bp) and H97 (CP015463.1:812033-812389, 356 bp). We designated the former as the H1 karyotype and the latter as the H2 karyotype in this study. In the 39Tr 2/5-2/4 marker analysis, 80 were heterokaryons (marked as H1H2 in Figure 4C) and 22 were homokaryons (marked as H1 or H2 in Figure 4C). The homokaryons of the H1 karyotype corresponded to the *A1* mating type, except for homokaryons Nos. 23 and 53 which were the *A2* mating type. Likewise, the H2 homokaryons were mostly the *A2* mating type, except for four homokaryons, Nos. 19, 54, 94, and 95, which were *A1* in the mating-type analysis. Additionally, the isolates No. 36 and No. 81, which were heterokaryons in the *A* mating-type marker analysis, were estimated to be homokaryons. The karyotype differences discovered here potentially result from transposon mobilization from H1 to H2. The H1 sequence contained a hAT-type non-autonomous DNA transposon homologous to ABR1, which was absent from H2 (Appendix A).

## 4. Discussion

The mating of basidiomycetes is normally governed by mating type genes clustered in two chromosomal mating type loci, *A* and *B*, resulting in tetrapolar mating behavior, whereas the mating of *A. bisporus* var. *bisporus* is controlled by the *A* mating type locus because of a loss of mating type function in the *B* locus [19]. Similar bipolar mating has been shown by other mushrooms, such as *Coprinellus disseminatus, Phanerochaete chrysosporium*, and *Pholiota nameko* [27,28,29]. In these mushrooms, the diversification of the *A* mating type locus is particularly important for successful reproduction in wild environments. 

The diversification of the *A* mating type locus has occurred in various ways, such as sequence variations in the *HD* genes, gene number variations in *HD* gene sets, and the insertion of DNA fragments into the mating type locus, depending on mushroom species. Sequence variations in the mating type genes are one way to generate mating type diversity, as shown in studies of *L. edodes* and *P. eryngii*, in which the sequence regions spanning the 5′-*HD1*–intergenic sequence–5′-*HD2* became hyper-variable [20,21]. The paralogous expansion and constriction of *HD* gene sets have been discovered in many mushrooms, such as *Coprinopsis cinerea* which carries three sets of *HD1*–*HD2* pairs in the archetypal *A* and diversifies through gene constriction [17,30]. The transposon insertion to the *A* mating type locus is the other form of *A* mating type evolution as in *A. bisporus* var. *burnettii* [19], *Neurospora* [31], and *Ustilago* [32]. Transposon insertion is also responsible for the emergence of the *A* mating type locus of *A. bisporus* var. *bisporus* H97 in this investigation (Figure 1 and Figure 2). The presence of multiple homologous sequences in various chromosomal DNAs (Figure 2) suggests that the *A* mating type locus may have acquired the insertion through inter-chromosomal transposition.

Lastly, transposon insertion enables the simple detection of the mating types (nuclear types) through PCR analysis. It is crucial to identify the nuclear type of the mycelial cells in *A. bisporus* for the separation of homokaryons from heterokaryons and for the verification of a specific nucleus in multi-nucleated heterokaryotic cells [10,13]. Numerous fingerprinting methods, such as RFLP, ISSR, and SSR, have been developed [11,12,15,16]; nevertheless, they are not simple or useful enough for routine application. However, as accessibility to genome sequence information increases, comparative analysis of these sequences has opened up new possibilities for discovering more precise and straightforward verification markers when combined with PCR. This has been applied in the karyotyping of *L. edodes* [20] and *P. eryngii* [21], and in the typing of mitochondrial DNA in *L. edodes* [33]. Our mating type marker study, conducted within the same framework as previous studies, illustrates that thorough analysis of genome information leads to the straightforward and reliable detection of karyotype which enables differentiation between homokaryons and heterokaryons and the evaluation of mating success in traditional mating and molecular breeding.

## Figures and Tables

**Figure 1 jof-09-00284-f001:**
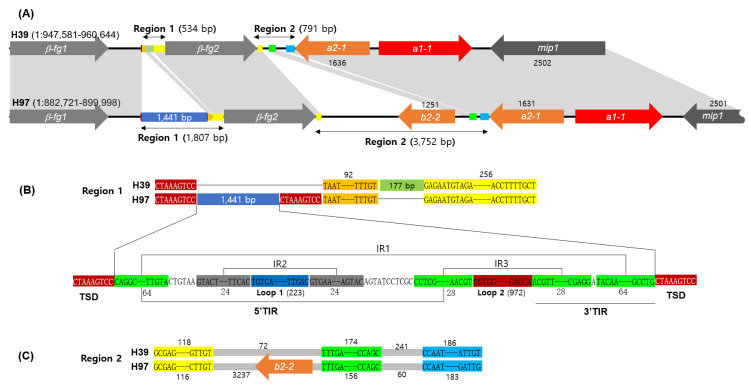
Comparison of the *A* mating types in *Agaricus bisporus* H39 and H97. (**A**) Gene arrangement of the *A* mating type loci located in chromosome 1. The shaded regions indicate the homologous sequence regions. The consensus sequences in Regions 1 and 2 are color-coded differentially. The numbers indicate the length of each gene or sequence region in the base pairs (bps). (**B**) Comparison of Region 1 in H39 with that of H97. The target site duplications (TSDs) are boxed in red, and the terminal inverted repeats (TIRs) are boxed in green. IR1, IR2, and IR3 are the inverted repeat sequences found inside the 1441 bp-long insertion in H97. (**C**) Comparison of Region 2 in H39 with that of H97. Region 2 has three consensus sequences, boxed in different colors.

**Figure 2 jof-09-00284-f002:**
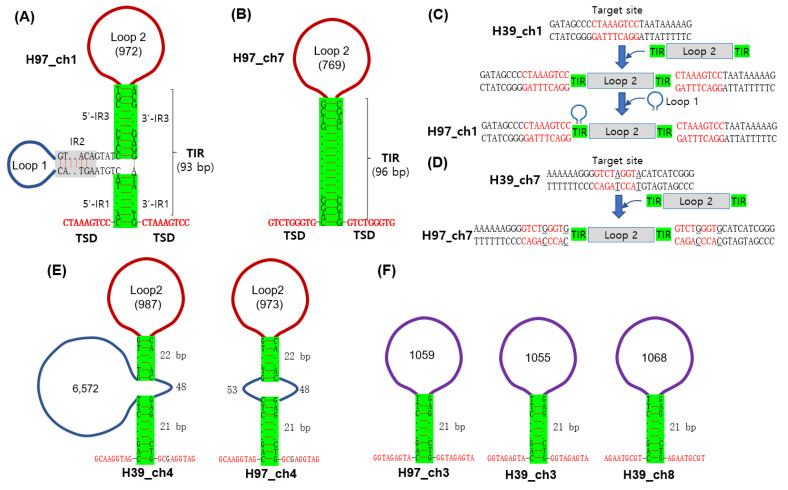
Transposon sequences found in different chromosomes in H39 and H97. The homologous sequences in the TIRs are boxed in green, and the TSDs are red-colored. (**A**) The structure of Region 1 in H97, composed of a serial connection of 5′TSD, 5′TIR, Loop 2, 3′TIR, and 3′TSD. 5′TIR has an additional insertion with IR2 and Loop 1, presumably another form of transposon. (**B**) The structure of the homologous transposon found at chromosome 7 of H97 (H97_ch7). The TIR and Loop 2 share high sequence homologies with those at chromosome 1 (H97_ch1). However, it is thought to be a more intact form since it lacks the Loop 1 insertion in the 5′TIR. (**C**) Possible explanation of the transposon insertion at the target site in the ancient chromosome. Region 1 in H97 depicted in (**A**) is thought to be constructed by sequential insertion of Loop 2 and Loop 1, whereas the target site in chromosome 1 of H39 remains intact. (**D**) Possible incorporation of the homologous transposon depicted in (**B**) to chromosome 7 of H97. The target site in H39 has two SNPs. (**E**) Homologous transposons found from chromosome 4 of H39 (H39_ch4) and H97 (9H97_ch4). They share high homology with the transposons depicted in (**A**,**B**), but have shown variations in the TIR regions. (**F**) Potential transposons with a 21 bp homologous sequence region in the TIR regions. They have homologous loop sequences but are not homologous to Loop 2. They share homology with the TIRs in (**A**,**B**) only in 21 bp from the TSD.

**Figure 3 jof-09-00284-f003:**
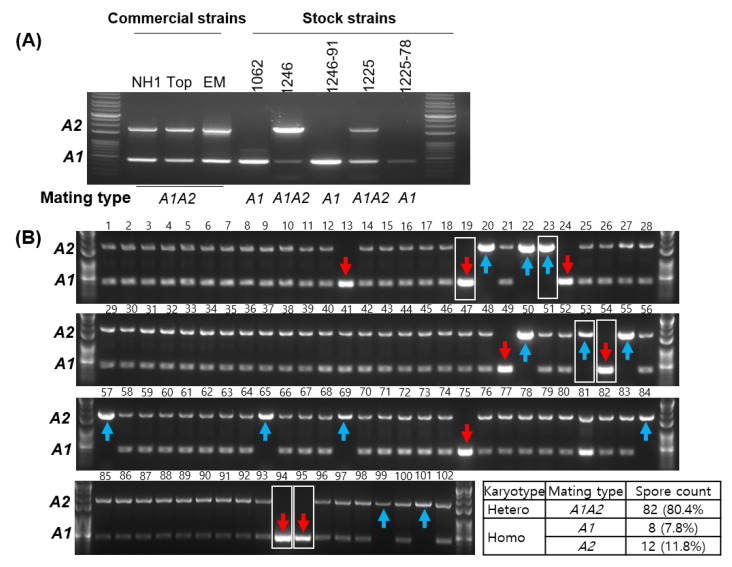
Identification of the *A* mating type of *Agaricus bisporus*. (**A**) Verification of the *A* mating types of commercial and stock strains. (**B**) Verification of the *A* mating types of the spore-borne mycelial isolates. The basidiospores, collected from the fruiting body of the NH1 strain, were plated on CDPM. The slow-growing mycelial colonies were isolated and their *A* mating types were investigated using PCR targeting of Region 1 in Figure 1. The homokaryons are indicated by arrows in different colors, where *A1*s are in red and *A2*s are in blue. Differently identified karyotypes from the *A* mating type marker analysis shown in Figure 4A are boxed. The distribution of the spore mating types is summarized in the table under the picture.

**Figure 4 jof-09-00284-f004:**
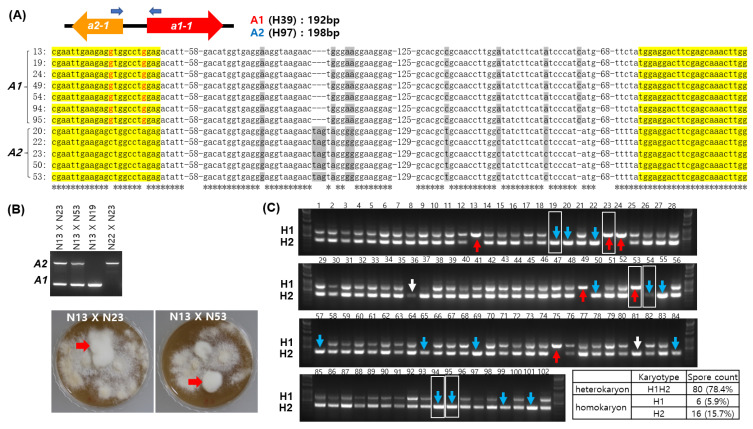
Validation of the karyotyping marker. (**A**) Sequence analysis of the *A* mating type regions in the spore-borne mycelial isolates. The sequence regions between *a2-1* and *a1-1* were determined after PCR. The consensus sequences in *a2-1* and *a1-1* are shaded in yellow and polymorphic sequences are shaded in grey. (**B**) Mating analysis between homokaryotic strains. Karyotyping was carried out after isolating heterokaryotic mates from the mating plate (red arrows). (**C**) Karyotyping of the spore-borne mycelial isolates by the 39Tr 2/5-2/4 marker [18]. The homokaryons are indicated by arrows in different colors, where H1s are in red and H2s are in blue. Differently identified karyotypes from the *A* mating type marker analysis shown in Figure 3B are boxed.

## Data Availability

Not applicable.

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
