# Peer review of "Structural Analysis of the A Mating Type Locus and Development of the Mating Type Marker of Agaricus bisporus var. bisporus"

_jof, 2023, doi:10.3390/jof9030284_

Round 1

Reviewer 1 Report

The authors have analyzed the mating type locus of the constituent nuclei (H39 and H97) of one of the first commercial strains of the button mushroom (Agaricus bisporus), Horst U1. This analysis showed that likely transposons have played a role in the diversification of the mating type between these nuclei. An interesting observation and useful to design a mating-specific PCR reaction.

A major concern is the part in the manuscript where the authors compare their mating-type markers with those used previously by Wei Gao (2013). As far as I know, the markers used by Wei Gao where were not designed to discriminate between H39 and H97 but designed to discriminate for mating type between an old traditional and 2 wild lines. A comparison between your primers and those of Wei Gao for discrimination between mating types of H39 and H97 are thus not valid. The authors should clarify this.

A few other remarks:

Line 22-23: “ It also enables.....homokaryons”. Better: “"This length difference enables the discrimination between homo- and heterokaryotic spores by PCR.”
Lines 25-26: “...which allows identification ....of A. bisporus.” This is only valid for the homokaryons of H39 and H97, probably not for mating types in other Homokaryons.
Lines 61-62: “moreover, ...dedikaryotization”. A number of research groups have done dekaryotisation of A. bisporus successfully. An example is Wei Gao, cited by the authors.
Lines 63-64: “The dikaryotization....protoplasts.”  Dedikaryotization is not done by plating dikaryotic protoplasts but by plating protoplasts derived from dikaryotic mycelium and select for monokaryotic regenerants. And reference 14 refers to protoplasting of basidia, not of mycelia.
Lines 113-114: It seems to me that the primers shown here are located only upstream of the mating locus.
Line 124: “...to exclude bi-nucleated mycelia.”  Better: “..to exclude heterokaryotic single spore cultures, since heterokaryons grow generally faster than homokaryons.
Line 271: “..occurred various ways..”. Change in “ ..occurred in various ways,..”.
Lines 291-295: Here the authors state that mating type markers based on sequence length difference are a simple and reliable way to discriminate homo- from heterokaryons.  This is a too general conclusion since the primers designed by the authors are useful for the homokaryons H39 and H97, but it is not known how the will be useful for other mating types (and thee are many in A. bisporus). The authors might rephrase their conclusion by stating that a thorough sequence analysis of mating types might in many cases allow the generation of mating-type specific markres.

Author Response

Responses to the reviewer 1

The authors have analyzed the mating type locus of the constituent nuclei (H39 and H97) of one of the first commercial strains of the button mushroom (Agaricus bisporus), Horst U1. This analysis showed that likely transposons have played a role in the diversification of the mating type between these nuclei. An interesting observation and useful to design a mating-specific PCR reaction.

A major concern is the part in the manuscript where the authors compare their mating-type markers with those used previously by Wei Gao (2013). As far as I know, the markers used by Wei Gao where were not designed to discriminate between H39 and H97 but designed to discriminate for mating type between an old traditional and 2 wild lines. A comparison between your primers and those of Wei Gao for discrimination between mating types of H39 and H97 are thus not valid. The authors should clarify this.

Response: Gao et al. (2013) demonstrated that four makers including G-6-PD, 39Tr2/5-2/4, PIN150, and P33N10 could discriminate homokaryons of certain traditional and wild lines. We tested the four markers to see if they are also effective in the discrimination of homokaryons of cultivated strains and some of stock strains and found only 39Tr2/5-2/4 yielded reliable result. Sequence of 39Tr2/5-2/4 is located at the chromosome 7 of H97 (CP015463.1:812033-812389, 356 bp) and H39 (CP015476.1:748009-748676, 668 bp). Therefore it does not represent the mating type because the A mating type locus is located at the chromosome 1. However, it was also effective in the discrimination of H97 and H39 as well as karyotyping of other cultivated strains. The difference between 39Tr2/5-2/4 and our A mating type marker is the former is for the identifying homokaryons while the latter is for the mating type of homokaryons. Both markers work well to the cultivated strains because of low genetic diversity within the cultivated strains. In order to avoid the misleading implication, we replaced the following line (L258-259) “These results indicate that the difference in length in Region 1 represents the A mating type better than the 39Tr 2/5-2/4 marker.” with “It should be noted that the 39Tr 2/5-2/4 marker was developed to discriminate the karyotype not the mating type.”

A few other remarks:

Line 22-23: “ It also enables.....homokaryons”. Better: “"This length difference enables the discrimination between homo- and heterokaryotic spores by PCR.”

Response: corrected as suggested.

Lines 25-26: “...which allows identification ....of A. bisporus.” This is only valid for the homokaryons of H39 and H97, probably not for mating types in other Homokaryons.

Response: Corrected as “The present study suggests that the A mating type locus in A. bisporus H97 has evolved through transposon insertion, allowing the discrimination of the mating type and thus the nuclear type between A. bisporus H97 and H39.”

Lines 61-62: “moreover, ...dedikaryotization”. A number of research groups have done dekaryotisation of A. bisporus successfully. An example is Wei Gao, cited by the authors.

Response: Yes indeed. Dedikaryotization has been employed for many mushrooms. We corrected the paragraph as follows:

“Moreover, A. bisporus can have up to 25 nuclei in a mycelial cell [13], presenting challenges particularly in molecular breeding where a modified nucleus has to be selected from the multiple unmodified nuclei within the targeted mycelial cell. Dedikaryotization provides a useful solution to this issue, allowing for the generation of homokaryons of A. bisporus through a simple process that involves plating protoplasts derived from heterokaryotic mycelium and selecting for homokaryotic regenerants [14]. Nevertheless, the generation and verification of homokaryons are still some of the main hurdles in the breeding of A. bisporus.”

Lines 63-64: “The dikaryotization....protoplasts.”  Dedikaryotization is not done by plating dikaryotic protoplasts but by plating protoplasts derived from dikaryotic mycelium and select for monokaryotic regenerants. And reference 14 refers to protoplasting of basidia, not of mycelia.

Response: The sentence was corrected as suggested (See above). The reference 14 isolated monokaryons both from protoplasting derived from the dikaryotic mycelia and from basidiospores which was described in their results section.

Lines 113-114: It seems to me that the primers shown here are located only upstream of the mating locus.

Response: The primers are indeed located at upstream region of HD genes. However we found the length difference in this region can represent the A mating type as shown by the sequence analysis in Fig. 4B.

Line 124: “...to exclude bi-nucleated mycelia.”  Better: “..to exclude heterokaryotic single spore cultures, since heterokaryons grow generally faster than homokaryons.

Response: Corrected as suggested.

Line 271: “..occurred various ways..”. Change in “ ..occurred in various ways,..”.

Response: corrected.

Lines 291-295: Here the authors state that mating type markers based on sequence length difference are a simple and reliable way to discriminate homo- from heterokaryons.  This is a too general conclusion since the primers designed by the authors are useful for the homokaryons H39 and H97, but it is not known how the will be useful for other mating types (and thee are many in A. bisporus). The authors might rephrase their conclusion by stating that a thorough sequence analysis of mating types might in many cases allow the generation of mating-type specific markers.

Response: We rephrased this paragraph with two additional citations as follow:

Numerous fingerprinting methods, such as RFLP, ISSR, and SSR, have been developed [11,12,15,16]; nevertheless, they are not simple or useful enough for routine use. But as accessibility to genome sequence information increases, comparative analysis of these sequences has opened up new possibilities for discovering more precise and straightforward verification markers when combined with PCR. This has been applied to the karyotyping of Lentinula edodes [20] and Pleurotus eryngii [21], and to the typing of mitochondrial DNA in L. edodes [33]. Our mating type marker study, conducted within the same framework as previous studies, illustrates that thorough analysis of genome information leads to the straightforward and reliable detection of karyotype, which enables differentiation between homokaryons and heterokaryons and the evaluation of mating success in traditional mating and molecular breeding.

  1. Kim, S.; Song, Y.; Ha, B.; Moon, Y.J.; Kim, M.; Ryu, H.; Ro, H.S. Variable number tandem repeats in the mitochondrial DNA of Lentinula edodesGenes2019, 10(7), 542.

Reviewer 2 Report

The manuscript describes an analysis of the mating type locus of A.bisporus. By comparing genes of two homokaryotic strains, the authors were able to detect and describe length differences in genes. These length differences could be used to differentiate between mating types in hetero and homo karyotic strains. The manuscript is generally well-written, but should be edited by a native English speaking.

L 13: delete “the”

L30: should be: most commonly cultivated edible..

L33: I don’t understand how global warming can be a problem for the mushroom industry?

Fig 3B and fig 4A seems to be the same. There is no reference to fig 3B, but I guess that in line 221, it should be Figure 3B?

L221 You selected slow growing mycelia, so the 19.6% homokaryons, has nothing to do with previous reported values which were mononucleate vs dinucleate spores. The 19.6% has to do with the author’s ability to pick slow growing mycelia. There is a risk that such figures will quoted in future papers.  

Author Response

Responses to the reviewer 2

The manuscript describes an analysis of the mating type locus of A.bisporus. By comparing genes of two homokaryotic strains, the authors were able to detect and describe length differences in genes. These length differences could be used to differentiate between mating types in hetero and homo karyotic strains. The manuscript is generally well-written, but should be edited by a native English speaking.

Response: The manuscript revised using a English Correction Service.

L 13: delete “the”

Response: Corrected as suggested

L30: should be: most commonly cultivated edible..

Response: Corrected as suggested

L33: I don’t understand how global warming can be a problem for the mushroom industry?

Response: Mushrooms are cultivated in a conditioned environment that requires keeping temperature in a certain range, usually lower than 25℃ for mycelial propagation and 20℃ for fruiting body development. It requires high energy cost particularly in summer season. Current global warming trend makes it even worse and therefore there is a strong demand for new strains that produce commercial quality fruiting bodies at elevated temperature to save energy cost.   

Fig 3B and fig 4A seems to be the same. There is no reference to fig 3B, but I guess that in line 221, it should be Figure 3B?

Response: Fig 3B and 4A are different in markers. Fig 3B use the A mating type marker whereas 4A is using the 39Tr 2/5-2/4 marker. Fig. 3B is mistakenly cited as Fig. 3A. We corrected this in the text.

L221: You selected slow growing mycelia, so the 19.6% homokaryons, has nothing to do with previous reported values which were mononucleate vs dinucleate spores. The 19.6% has to do with the author’s ability to pick slow growing mycelia. There is a risk that such figures will quoted in future papers.  

Response: There is misunderstanding on this result. The slow growing mycelia were regenerated from the germinating spores. The mycelia from homokaryotic (monokaryotic) spores grow slower than those from the heterokaryotic (dikaryotic) spores. The 3%-rate of homokaryons reported in the references11 and12 is the result of random selection of spore-bourne mycelia.   

Round 2

Reviewer 1 Report

Paragraph 3.4 (lines 240-266) still contains a comparison of the PCR markers of the authors and those of Gao et al. Gao, however, designed these primers for other homokaryons and for regions on chromosome 7, thus not for mating type markers. A comparison of results between primers used for complete different puposes does not make sense.

On lines 121-122 authors should state that the primes were used to identify variable region 1 of the mating types.

Author Response

We'd like thank to your insightful comments.  In response to your feedback, we made revisions in the hopes that they meet your standards.

1) We agree on your concern on the direct comparison between two different markers. Accordingly, we rewrote the Section 3.4. In the revised version we validated the A mating type marker with sequence analysis (Figure 4A) and mating experiment (Figure 4B). Then we compared the current mating type marker with the 39Tr 2/5-2/4 marker in identifying karyotypes of the spore-borne mycelia (Figure 4C). In the revised manuscript, we assigned the H39 type homokaryons as H1 karyotype and the H97 type as H2 karyotype to discriminate them with the A mating types. Figure 4 was rearranged accordingly with the main text and Supplementary Figure S4 was additionally included to explain the presence of a transposon in H1 of the 39Tr 2/5-2/4 marker. The revised paragraphs are as follows:

"Validation of our A mating type marker was performed by direct sequencing of the variable sequence region located between the HD domain regions of a1-1 and a2-1 in the A mating type loci as previously described [20]. Our sequence analysis on some of the homokaryons in Figure 3B revealed that the A1 homokaryons shared identical sequences, while the A2 homokaryons shared the same sequence polymorphism, as shown in Figure 4A. Validity of the A mating type marker was further demonstrated by successful mating of Nos. 23 and 24 with No. 13 and failure in mating between Nos. 13 and 19, as well as Nos. 22 and 23 (Figure 4B).

The karyotypes of the spore-borne mycelia shown in Figure 3B were compared to those determined using the 39Tr 2/5-2/4 marker, which was developed by Gao et al. [18]. The 39Tr 2/5-2/4 marker is located on chromosome 7 of H39 (CP015476.1:748009-748677, 668 bp) and H97 (CP015463.1:812033-812389, 356 bp). We designated the former as the H1 karyotype and the latter as the H2 karyotype in this study. In the 39Tr 2/5-2/4 marker analysis, 80 were heterokaryons (marked as H1H2 in Figure 4C) and 22 were homokaryons (marked as H1 or H2 in Figure 4C). The homokaryons of the H1 karyotype corresponded to the A1 mating type, except for the homokaryons Nos. 23 and 53, which were A2 in the mating type. Likewise, the H2 homokaryons were mostly A2 mating type, except for four homokaryons, Nos. 19, 54, 94, and 95, which were A1 in the mating type analysis. Additionally, the isolates No. 36 and No. 81, which were heterokaryons in the A mating type marker analysis, were estimated to be homokaryons. The karyotype differences discovered here potentially result from transposon mobilization from H1 to H2. The H1 sequence contained a hAT-type non-autonomous DNA transposon homologous to ABR1, which was absent from H2 (Supplementary Figure 4)."

2) We restated the sentence as follows:

"Identification of the mating type was achieved by a targeted PCR with a primer set (5’-ATCAGTACTTTGAAGCGTGGTAG-3’ and 5’-GTGCTCATACCTCACAATCACTG-3’) specific to the variable sequence in the Region 1 of the A mating type locus. "

3) The main text was corrected thoroughly by MDPI English service.